# Molecular Prevalence of *Blastocystis* sp. from Patients with Diarrhea in the Republic of Korea

**DOI:** 10.3390/microorganisms12030523

**Published:** 2024-03-05

**Authors:** Ji-Young Kwon, Jong-Hoon Choi, Hee-Il Lee, Jung-Won Ju, Myoung-Ro Lee

**Affiliations:** Division of Vectors and Parasitic Diseases, Korea Disease Control and Prevention Agency, 187 Osongsaenmyeong 2-ro, Osong-eup, Heungdeok-gu, Cheongju 28159, Chungcheongbuk-do, Republic of Korea; kjiy31@korea.kr (J.-Y.K.); choijhoon@korea.kr (J.-H.C.); isak@korea.kr (H.-I.L.); jupapa@korea.kr (J.-W.J.)

**Keywords:** *Blastocystis* sp., Republic of Korea, diarrhea patients, subtype, allele

## Abstract

*Blastocystis* sp. is the most common intestinal protozoan affecting human health worldwide. Several studies have reported the prevalence of *Blastocystis* sp. in various regions of the Republic of Korea. However, limited data are available on the prevalence and subtype (ST) distribution of this parasite among regions. Therefore, we investigated the prevalence and ST distributions of this parasite in the Republic of Korea. For this purpose, 894 stool specimens were collected from patients with diarrhea and tested for the presence of *Blastocystis* sp. using PCR analysis. The isolates were subsequently subtyped. The overall prevalence was 11.6%. Of the 104 isolates, ST3 was the most prevalent, followed by ST1. Additionally, a single case of the rare subtype ST8 was identified, representing the first reported case in the Republic of Korea. The results suggested that the predominance of ST3 observed in this study reflects human-to-human transmission with low genetic diversity within the ST, while ST1 transmission is likely correlated with animals. In the future, to better understand *Blastocystis* sp. transmission dynamics, human, animal, and environmental factors should be studied from a “One Health” perspective.

## 1. Introduction

*Blastocystis* sp. is a protozoan parasite that resides within the gastrointestinal tract of a wide range of hosts, including mammals, birds, fish, reptiles, insects, and humans [1,2]. This enteric microorganism is a causative agent of waterborne and foodborne diseases. The infective cysts are transmitted through the fecal–oral route [3,4]. An estimated more than 1 billion people are infected with this parasite worldwide, with a prevalence of 10–15% in developed countries and >50% in developing nations [5]. Although most infected people are asymptomatic carriers, some vulnerable individuals develop symptoms, including diarrhea, abdominal pain, gas, and nausea, as well as extraintestinal symptoms such as urticaria and rash [6,7,8,9,10]. *Blastocystis* sp. colonization is reportedly associated with a healthy gut microbiome rather than intestinal dysbacteriosis [6]. However, in vitro and in vivo studies combined with genomic analyses have identified various virulence factors that may be related to the genetic variation in *Blastocystis* isolates [2,11].

The genetic variation in *Blastocystis* sp. in birds and mammals (including humans) is categorized into at least 28 subtypes (STs), including ST1–ST17, ST21, and ST23–ST32, based on polymorphisms in the small subunit of the rRNA (SSU rRNA) gene [12]. Among these, ST1–10, ST12, ST14, ST16, and ST23 have been reported in humans, with ST1–ST4 representing >90% of subtyped human isolates [13,14]. Accordingly, most ST1–ST4 infections are transmitted via human-to-human routes, while the other STs predominate in certain host groups, such as mammals or birds, and are transmitted through human–animal pathways [15,16].

Genetic variant studies using Multilocus Sequence Typing (MLST) schemes have revealed intra-genetic variation among the STs [14,17]. More specifically, following isolation from symptomatic patients, *Blastocystis* sp. subtypes and 18S alleles have been reported in several countries [9,18,19,20,21]. In Iran, the predominant *Blastocystis* subtypes responsible for causing diarrhea, bloating, nausea, and urticaria are ST1 (allele 4), ST2 (alleles 9, 11, and 12), ST3 (alleles 34, 37, and 52), and ST6 (allele 122) [19,20]. In Colombia, those causing diarrhea were classified as ST2 (allele 9), and those associated with irritable bowel syndrome (IBS) were classified as ST3 (allele 34) [21].

Although studies in the Republic of Korea (ROK) have reported the STs of *Blastocystis* sp. [22,23,24,25], 18S alleles have not been reported for *Blastocystis* STs isolated from humans. Therefore, the primary aim of the current study was to evaluate the genetic variation and geographical distribution of subtypes and 18S alleles in patients with diarrhea in the ROK.

## 2. Materials and Methods

### 2.1. Sample Collection and DNA Extraction

In total, 894 stool samples were collected from patients with diarrhea via the Enteric Pathogens Active Surveillance Network (Enter-Net) of the Korea Disease Control and Prevention Agency (KDCA) in 2022. Enter-Net is organized by the KDCA and comprised of 11 local Public Health Institutes and 73 participating hospitals. Samples were collected from patients suspected of waterborne and foodborne infectious diseases to isolate and identify the causative agent. The samples were collected from the following 11 regions in the ROK with 10 mL screw-cap tubes: Incheon, Seoul, Chungcheong buk-do, Gyeongsangbuk-do, Daejeon, Daegu, Jeollabuk-do, Gyeongsangnam-do, Busan, Jeollanam-do, and Jeju Island (Table 1). The samples were stored at −20 °C before DNA extraction. Total DNA was extracted from 250 mg (or 250 µL for watery samples) of fecal samples using the QIAcube HT apparatus and the DNeasy^®^ PowerSoil^®^ Pro Kit (QIAGEN, Hilden, Germany), following the manufacturer’s protocol. To enhance the extraction efficacy, the FastPrep-24™ 5G (MP Biomedicals, Solon, OH, USA) was employed for physical disruption at 6 m/s for 40 s. The extracted DNA was preserved at −20 °C until PCR analysis. The samples were processed to detect viruses, bacteria, and parasitic protozoa.

Ethical approval was not required because this study managed to assess public welfare via a fact-finding survey (Infectious Disease Control and Prevention Act).

### 2.2. PCR Amplification

PCR amplification of *Blastocystis* sp. was carried out using barcoding region primers BhRDr (5′-GAGCTTTTTAACTGCAACAACG-3′) and RD5 (5′-ATCTGGTTGATCCTGCCAGT-3′) for SSU rRNA [26]. PCR amplification included the following steps: initial denaturing at 94 °C for 5 min and 35 cycles of 94 °C for 1 min, 59 °C for 1 min, and 1 min at 72 °C; a final extension at 72 °C was performed for 2 min. A negative control was used to detect contamination in each reaction. The amplified products (600 bp specific band) were visualized using a Qiagen QIAxcel Capillary Electrophoresis system (QIAGEN, Hilden, Germany).

### 2.3. DNA Sequencing Analysis

All positive PCR products were purified and sequenced. Nucleotide sequences were analyzed and trimmed using BioEdit version 7.2.5 (Ibis Therapeutics Inc., Carlsbad, CA, USA). The resulting nucleotide sequences were subjected to BLAST searches (https://blast.ncbi.nlm.nih.gov/Blast.cgi, accessed on 20 August 2023) using the data available on GenBank (http://www.ncbinlm.nih.gov/genbank/, accessed on 20 August 2023). STs were determined based on an exact match or an identity ≥99% against all known *Blastocystis* STs, with a query coverage of ≥99%.

### 2.4. Phylogenetic Analysis and Allele Discrimination

Multiple sequence alignments were performed using ClustalW BioEdit version 7.2.5 (Ibis Therapeutics Inc., Carlsbad, CA, USA). The Mega Align program (DNASTAR, Madison, WI, USA) was used to determine the similarity and difference rates between sequences. Finally, phylogenetic analysis was performed with MEGA software (version 5.02) using neighbor-joining analysis with a Kimura 2-parameter model estimated using bootstrap analysis with 1000 replicates. Each sequence was identified based on the accession number, host, origin, and genotype. All established sequences were submitted to the MLST database (https://pubmlst.org/blastocystis/, accessed on 20 August 2023) for ST corroboration and relevant allele identification [26].

### 2.5. Statistical Analysis

The relationships between *Blastocystis* sp. and sex or age groups were assessed using the Chi-square (X^2^) test or Fisher’s exact test. A *p* value < 0.05 was considered significant. All statistical analyses were performed using GraphPad Prism 5 software 5.01.

## 3. Results

### 3.1. Regional Distribution of Blastocystis sp.

In total, 894 samples were collected from patients with diarrhea in this study, 104 of which were positive for *Blastocystis* sp. (Table 1). Busan had the highest number of positive cases, followed by Gyeongsangnam-do and Seoul. Three different *Blastocystis* sp. STs were identified, including ST1 (11/104, 10.6%), ST3 (92/104, 88.5%), and ST8 (1/104, 1.0%). ST3 (allele34) was found to be the most common *Blastocystis* sp., which was detected in all sample collection areas, but ST3 (allele36) was detected only in five areas: Seoul, Gyeongsangbuk-do, Daegu, Gyeongsangnam-do, and Jeju Island. The prevalence of *Blastocystis* sp. ST1 was identified in Chungcheongbuk-do, Daejeon, Daegu, Gyeongsangbuk-do, Gyeongsangnam-do, Jeollabuk-do, and Jeju Island (Figure 1). Of the 104 *Blastocystis* sp. cases, 18 were confirmed to be co-infected with *Bacillus cereus*, *Clostridium perfringens*, *Escherichia coli*, *Salmonella* spp., and *Shigella* spp. Mixed ST infections with *Blastocystis* sp. were not detected.

### 3.2. Prevalence of Blastocystis sp.

*Blastocystis* sp. was detected in 45 male (45/461, 9.8%) and 59 female (59/433, 13.6%) patients. However, infection was not associated with sex (*p* = 0.118, odds ratio = 1.395, 95% confidence interval = 0.9247–2.106; Table 2).

The ages of the patients ranged from 1 month to 97 years (mean: 49 years; SD: ±30 years). The patients were divided into eight categories based on age; *Blastocystis* sp. was detected in all age groups. The largest number of positive cases was detected within the 0–9 age group, with the associated infection rate significantly higher in the other age groups (Chi-square = 23.09, *p* = 0.001; Table 2).

### 3.3. Phylogenetic Analysis

All 104 *Blastocystis*-positive samples were successfully sequenced. The ST1 sequences (KDCA 56, 57, 59, and 93) were closely related to the ST1 sequence (MK922974) isolated from humans in China. In addition, ST1 sequences (KDCA 17, 29, 80, and 104) were similar to an ST1 sequence (MH997830) obtained from pigs in the ROK. The ST3 sequence showed a close genetic relationship with human ST3 sequences from Spain, India, and China. The ST8 sequence (KDCA 64) obtained in this study was identical to those from humans in Brazil (MN585850), Spain (MN836842), and the USA (EU679348), as well as those from monkeys in Colombia (OP329411) (Figure 2). Excluding the previously registered sequences, all sequences obtained in this study were deposited into GenBank under accession number OR447547.

### 3.4. Distribution of 18S Alleles and Genotypes

Allele analysis of the *Blastocystis* SSU rRNA gene revealed four variants in the positive samples. Based on the *Blastocystis* 18S database, the detected alleles were identified as allele 4 (100%; 11/11) for ST1, alleles 34 (90.4%; 83/92) and 36 (9.6%; 9/92) for ST3, and allele 21 (100%; 1/1) for ST8 (Figure 3). To evaluate the intra-ST diversity and identify genotypes, partial SSU rDNA gene sequences belonging to ST1 or ST3 were aligned (Figure 4). This comparison identified three ST1 (ST1-1 to ST1-3) and two ST3 (ST3-1 and ST3-2) genotypes in patients with diarrhea in the ROK. All ST1 isolates (ST1-1 to ST1-3) belonged to allele 4, with one or four variable positions. Meanwhile, ST3-1 belonged to allele 34, and ST3-2 to allele 36, with a single variable position.

## 4. Discussion

*Blastocystis* is one of the most common genera of intestinal eukaryotes found in humans worldwide [16]. The prevalence of *Blastocystis* sp. reportedly ranges from 0.5% in Thailand to 100% in Senegal [12,27]. Several Asian studies have determined the average prevalence of *Blastocystis* sp. in China (11.62%), Japan (1.0%), Malaysia (19.25%), Nepal (25.2%), Singapore (3.3%), the Philippines (49.1%), and Indonesia (34.25%) [16,28]. The prevalence of *Blastocystis* sp. is particularly high within marginalized areas in terms of public health, industrialization, and socioeconomic status [29]. In China, the prevalence of *Blastocystis* sp. reaches 32.6% in less industrialized and socio-economically marginalized areas, whereas in industrialized urban areas, it is as low as 1.9% [30]. Similarly, in Malaysia, low household income and untreated tap water use were identified as important risk factors for *Blastocystis* sp. infection [31]. In Nepal, infection is widespread in rural communities due to the consumption of untreated water [32]. Meanwhile, in the Philippines, the prevalence of *Blastocystis* sp. infection was high among pet owners [33]. The high prevalence was found to have poor hygiene and close contact between humans and animals in the area surveyed [34]. In contrast, Japan reported a decrease from the previously reported prevalence (0.5–2.5%) due to improved public health [28]. Comparatively, Singapore is an urbanized city-state with a low prevalence rate [35]. In previous studies, the prevalence of *Blastocystis* sp. in the ROK was 5.6–9.2%, while the current study detected a rate of 11.6% [22,23,24,25]. This discrepancy may stem from the differences in sample size and breadth of sample collection locations. That is, the current study included more than 500 stool samples collected across various regions and age groups. In terms of distribution by age, the prevalence of *Blastocystis* sp. was high in those aged 0–9 and >60 years, presumably due to relatively weakened immunity compared with the other groups [36].

One of the most common symptoms associated with *Blastocystis* sp. is diarrhea, with several studies reporting a correlation between diarrhea and specific subtypes. For example, in Iran, ST1, ST2, ST3, and ST6 were detected in symptomatic patients, while ST2 and ST3 were detected in Colombian patients with diarrhea [19,20,21]. Similarly, a Saudi Arabian study reported that ST3 predominated in patients with abdominal pain and diarrhea [37]. However, ST1, ST2, and ST3 have also been reported in asymptomatic carriers. Hence, further investigation into the pathological role of *Blastocystis* subtypes is needed.

As previously reported, ST3 of *Blastocystis* sp. is the predominant subtype globally [14,38]. This aligned with the findings of the current study, with ST3 as the dominant subtype in the ROK, followed by ST1. Our phylogenetic analysis revealed that the ST3 sequence shared sequence similarity with other human isolates from around the world, whereas no similar sequences have been reported in domestic animals. Hence, ST3 is transmitted via human-to-human routes. Moreover, ST3 was detected in all study areas, suggesting its broad distribution among humans in the ROK. ST1 is the predominant subtype isolated from humans [14,39]. In the current study, most ST1 sequences were phylogenetically similar to others reportedly isolated from humans. Meanwhile, certain ST1 sequences were closely related to ST1 sequences isolated in domestic pigs (MH997827). This suggests indirect zoonotic transmission caused by exposure to contaminated water, food, or aerosols [38,40]. In addition, although ST2 has been previously reported in the ROK [22], it was not detected in the current study. Meanwhile, we identified *Blastocystis* ST8 in the ROK for the first time. ST8 is typically isolated from non-human primates and rarely humans [15]. Although it has been reported in animals and contaminated water sources in Asia, it has not been previously reported in East Asian populations [41]. In fact, *Blastocystis* ST8 has only been previously isolated in humans within the Americas and Europe [38]. The nucleotide sequence of ST8 detected in this study was 100% consistent with that of the American and European strains. Although the patient’s overseas travel history could not be ascertained, overseas inflow was suspected.

DNA barcoding is a useful method for determining intra-ST diversity [21]. According to allelic discrimination related to the identified subtypes in humans, globally, the most frequent alleles in each ST are as follows: ST1 (alleles 4 and 2), ST2 (allele 9), ST3 (alleles 34, 36, and 37), ST4 (alleles 42 and 91), and ST8 (allele 21) [2,42]. Genetic analysis in this study revealed that all four alleles detected in the *Blastocystis* sp. cross-corresponded with previously reported alleles. Analysis of the variable positions within the alignment of each ST revealed three ST1 genotypes in 11 isolates. In addition, two ST3 genotypes were identified. Although ST3 was detected most frequently, its genetic diversity was low. Considering that ST3 has never been reported in livestock or farm animals, the low genetic diversity could be associated with it being transmitted exclusively via human-to-human routes in the ROK. Among the ST genotypes, ST3-1 (79.8% of all ST isolates) predominated, representing the main cause of *Blastocystis* sp. transmission in the ROK.

Certain limitations were noted in the current study. No data were available on symptoms other than diarrhea, nor was information related to environmental or animal contact provided. In the future, multicenter epidemiological studies are needed to evaluate the role of environmental factors, animal contact, and overseas travel history to confirm the active circulation of *Blastocystis* sp. in the ROK. In addition, studies from the “One Health” perspective would help develop a more comprehensive understanding of *Blastocystis* sp. transmission and prevalence.

## Figures and Tables

**Figure 1 microorganisms-12-00523-f001:**
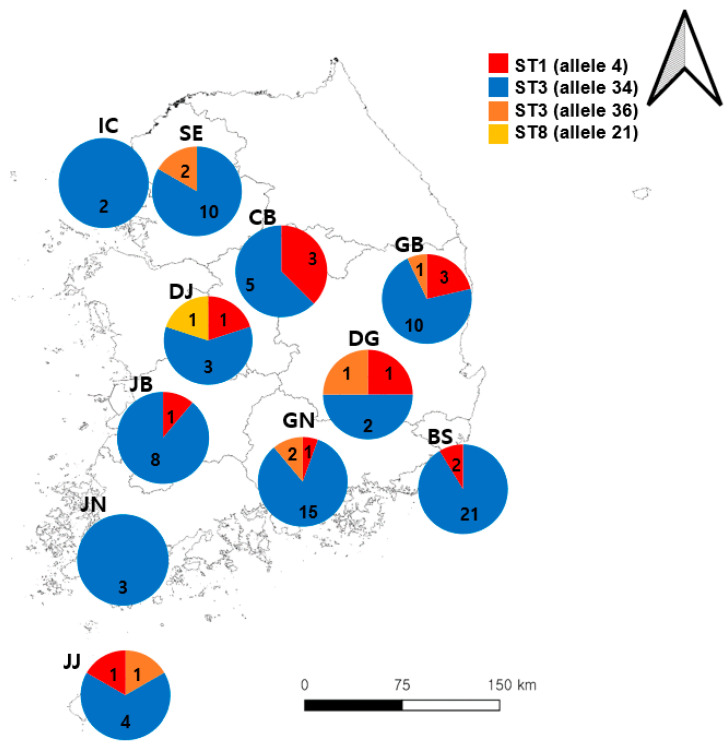
Geographical frequency of *Blastocystis* sp. subtypes (STs) detected in 104 human fecal samples from the Republic of Korea (ROK). The pie charts represent the proportion of subtypes (alleles) detected in each area; the Arabic numerals represent the number of cases detected. Abbreviations: BS, Busan; CB, Chungcheongbuk-do; DG, Daegu; DJ, Daejeon; GB, Gyeongsangbuk-do; GN, Gyeongsangnam-do; IC, Incheon; JB, Jeollabuk-do; JJ, Jeju Island; JN, Jeollanam-do; SE, Seoul.

**Figure 2 microorganisms-12-00523-f002:**
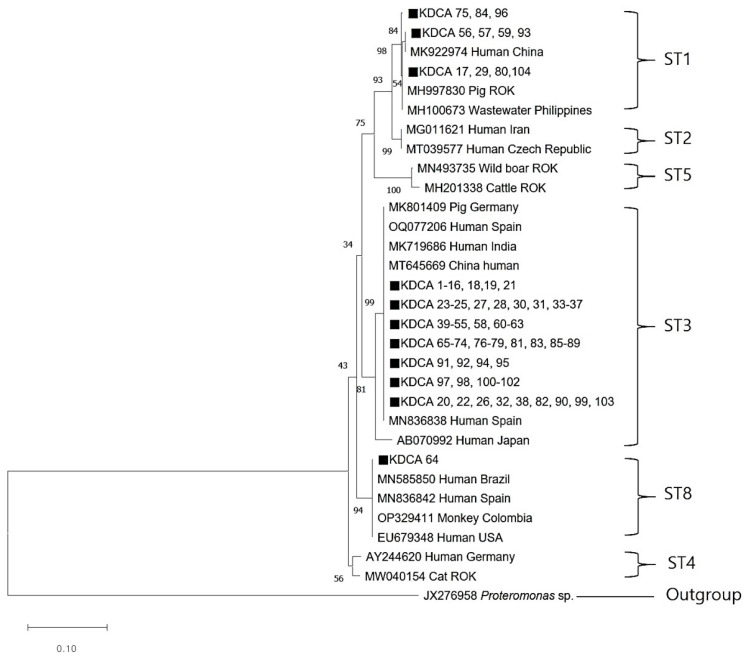
Phylogenetic relationships between *Blastocystis* sp. The relationships between the genotypes of *Blastocystis* sp. identified in this study and known genotypes published in GenBank were inferred using a neighbor-joining analysis of SSU rRNA gene sequences based on genetic distances calculated by the Kimura 2-parameter model. The numbers on the branches represent percent bootstrapping values from 1000 replicates. The closed black squares represent each genotype identified in this study. Scale bar = 0.10 replacements per nucleotide site.

**Figure 3 microorganisms-12-00523-f003:**
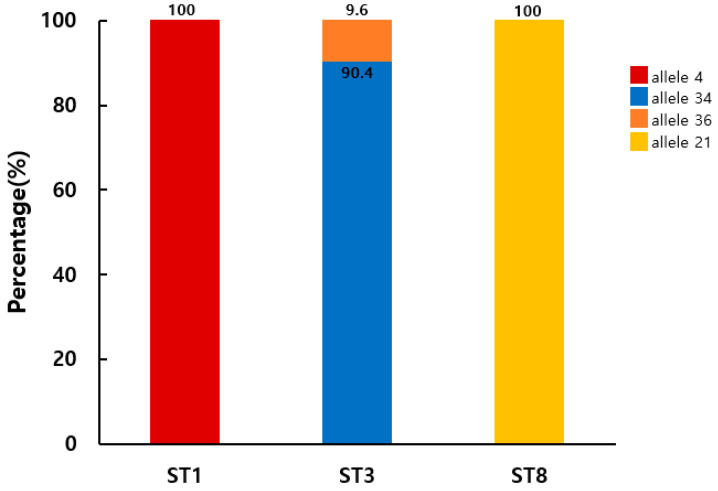
Distribution and frequency of *Blastocystis* sp. SSU rRNA alleles among subtypes isolated from 104 stool samples of patients in the Republic of Korea.

**Figure 4 microorganisms-12-00523-f004:**
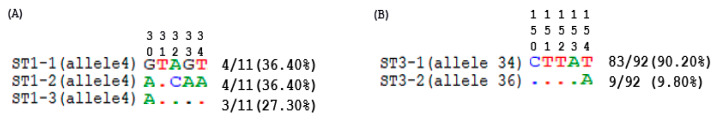
Alignment of partial SSU rRNA gene sequences from *Blastocystis* sp. ST1 (**A**) and ST3 (**B**) isolates. The variable positions for the reference sequences (genotype ST1-1, ST3-1) are indicated above the alignment (vertical number). Within each ST, the identified genotypes are indicated on the left side of the alignment, and the corresponding reference nucleotide sequences are indicated by dashes. On the right side of each alignment are the total number and percentage of isolates identified in this study.

**Table 1 microorganisms-12-00523-t001:** *Blastocystis* sp. detection in fecal samples from patients with diarrhea in 11 regions of the Republic of Korea.

Region (Abbreviation)	Samples (*n*)	*Blastocystis* sp.-Positive				
		Positive Samples (*n*)	Prevalence (%)	ST1 (*n*)	ST3 (*n*)	ST8 (*n*)
Incheon (IC)	35	2	5.7	-	2	-
Seoul (SE)	66	12	18.2	-	12	-
Chungcheongbuk-do (CB)	94	8	8.5	3	5	-
Gyeongsangbuk-do (GB)	98	14	14.3	3	11	-
Daejeon (DJ)	47	5	10.6	1	3	1
Daegu (DG)	118	4	3.4	1	3	-
Jeollabuk-do (JB)	100	9	9.0	1	8	-
Gyeongsangnam-do (GN)	99	18	18.2	1	17	-
Busan (BS)	106	23	21.7	-	23	-
Jeollanam-do (JN)	37	3	8.1	-	3	-
Jeju Island (JJ)	94	6	6.4	1	5	-
Total	894	104	11.6	11	92	1

ST, subtype.

**Table 2 microorganisms-12-00523-t002:** Distribution of *Blastocystis* sp. infection among patients with diarrhea in the Republic of Korea.

Variable	Samples (*n*)	*Blastocystis* sp.-Positive					
		Positive Samples (*n*)	Prevalence (%)	ST1 (*n*)	ST3 (*n*)	ST8 (*n*)	*p* Value
Number of samples	894	104	11.6	11	92	1	
Sex							
Male	461	46	10.0	5	41	-	0.118
Female	433	58	13.4	6	51	1	
Age group (years)							
0–9	188	39	20.7	1	38	-	0.001
10–19	54	3	5.6	-	3	-	
20–29	32	4	12.5	-	4	-	
30–39	30	3	10.0	-	3	-	
40–49	48	4	8.3	-	4	-	
50–59	97	7	7.2	1	5	1	
60–69	149	10	6.7	2	8	-	
≥70	296	34	11.5	7	27	-	

## Data Availability

Patient consent was not required for this study as it involved a retrospective analysis of specimens collected for legally stipulated research survey projects, and no further investigations were conducted with the submitted specimens.

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
