# Peer review of "Molecular Prevalence of Blastocystis sp. from Patients with Diarrhea in the Republic of Korea"

_microorganisms, 2024, doi:10.3390/microorganisms12030523_

Round 1

Reviewer 1 Report

Comments and Suggestions for Authors

Overall, I am so excited while reading this report as it is effectively communicates its findings and contributes valuable insights to the understanding of Blastocystis sp. epidemiology and subtype distribution in South Korea. The limitations are acknowledged, and directions for future research are suggested, indicating areas for further investigation to deepen understanding of this protozoan's transmission dynamics and public health impact. However, for improving this report, consider the following points:

- Broader Implications of Findings: While the study focuses on the prevalence and subtype distribution of Blastocystis sp. in South Korea, the discussion could be expanded to address the broader implications of these findings on global public health based on the available literature, particularly in regions with similar environmental and socio-economic conditions. This would make the study more relevant to an international audience.

- Comparison with Other Studies: The report could benefit from a more detailed comparison with other studies, especially those conducted in different geographical regions or among different populations. Such comparisons could help in understanding the global diversity of Blastocystis sp. and its implications for public health.

-In-depth Analysis of Subtype Pathogenicity: Although the study mentions the presence of various subtypes, it could delve deeper into the potential differences in pathogenicity or clinical significance among these subtypes in the discussion section. Providing a more detailed analysis of how different subtypes may affect human health could enhance the manuscript's impact based on the available literature.

- Methodological Details: While the methods are adequately described, providing more details on the selection criteria for participants, the rationale behind the sample size, and the statistical methods used for analyzing the data could improve the transparency and reproducibility of the study.

- Addressing Limitations More Comprehensively: The report acknowledges certain limitations, but it could further discuss the implications of these limitations on the study's findings and how they could be addressed in future research. For example, exploring methods to differentiate between colonization and infection could provide more insight into the clinical relevance of Blastocystis sp. subtypes.

- Future Research Directions: While future research directions are mentioned, the manuscript could provide more specific recommendations for future studies, such as investigating the role of environmental factors in the transmission of Blastocystis sp. or exploring the effectiveness of potential interventions.

Reviewer 2 Report

Comments and Suggestions for Authors

In this manuscript, Kwon et al. sequenced a polymorphic region of the small subunit of the ribosomal RNA gene from Blastocystis sp., to characterize the prevalence, diversity, and regional distribution of this intestinal parasite in patients with diarrhea, from eleven regions in the Republic of Korea. 894 stool samples were analyzed; of these, 104 tested positive for Blastocystis sp. The highest prevalence was in patients under 10 years. Multiple subtypes were identified and a phylogenetic analysis of the subtypes was also performed.

The study is clearly described and the data were correctly analyzed. This work should be relevant to physicians (mostly gastroenterologists) and policymakers from the Republic of Korea (and possibly other countries), regarding the impact of Blastocystis sp. infection in the population.

Minor remarks to the authors:

1. Throughout the manuscript, please check whether "18s" should be written as "18S".

2. In lines 101-102, the sentence "Fisher’s exact test or Chi-square (χ2 ) was used to assess the association sex or age group." is not fully clear. Please rephrase "(...) the association sex or age group".

3. Please provide Figure 2 in higher resolution, as the bootstrapping values in the upper branches are not readable.

4. In the legend of Figure 2, please explain the meaning of the scale bar labeled with 0.10 at the bottom of this figure.

6. In the discussion, please indicate the limitations of the study.

7. In the data availability statement, please mention if the sequence data generated in this study are publicly available (if so, please indicate the repository and accession number) or if they are available from the corresponding author upon reasonable request.
